# The Valproic Acid Derivative Valpromide Inhibits Pseudorabies Virus Infection in Swine Epithelial and Mouse Neuroblastoma Cell Lines

**DOI:** 10.3390/v13122522

**Published:** 2021-12-15

**Authors:** Sabina Andreu, Inés Ripa, Beatriz Praena, José Antonio López-Guerrero, Raquel Bello-Morales

**Affiliations:** 1Departamento de Biología Molecular, Universidad Autónoma de Madrid, Cantoblanco, 28049 Madrid, Spain; ines.ripa@cbm.csic.es (I.R.); jal@cbm.csic.es (J.A.L.-G.); raquel.bello-morales@uam.es (R.B.-M.); 2Centro de Biología Molecular Severo Ochoa, Spanish National Research Council—Universidad Autónoma de Madrid (CSIC-UAM), Cantoblanco, 28049 Madrid, Spain; 3MU Bond Life Sciences Center, University of Missouri, Columbia, MO 65211-7310, USA; bpk9f@missouri.edu

**Keywords:** valpromide, pseudorabies virus, PRV, suid herpesvirus 1, SuHV-1, antiviral, herpesvirus

## Abstract

Pseudorabies virus (PRV) infection of swine can produce Aujeszky’s disease, which causes neurological, respiratory, and reproductive symptoms, leading to significant economic losses in the swine industry. Although humans are not the natural hosts of PRV, cases of human encephalitis and endophthalmitis caused by PRV infection have been reported between animals and workers. Currently, a lack of specific treatments and the emergence of new PRV strains against which existing vaccines do not protect makes the search for effective antiviral drugs essential. As an alternative to traditional nucleoside analogues such as acyclovir (ACV), we studied the antiviral effect of valpromide (VPD), a compound derived from valproic acid, against PRV infection in the PK15 swine cell line and the neuroblastoma cell line Neuro-2a. First, the cytotoxicity of ACV and VPD in cells was compared, demonstrating that neither compound was cytotoxic at a specific concentration range after 24 h exposure. Furthermore, the lack of direct virucidal effect of VPD outside of an infected cell environment was demonstrated. Finally, VPD was shown to have an antiviral effect on the viral production of two strains of pseudorabies virus (wild type NIA-3 and recombinant PRV-XGF) at the concentrations ranging from 0.5 to 1.5 mM, suggesting that VPD could be a suitable alternative to nucleoside analogues as an antiherpetic drug against Aujeszky’s disease.

## 1. Introduction

Pseudorabies virus (PRV), also known as suid herpes virus type 1 (SuHV-1), is a pathogen that belongs to the subfamily *Alphaherpesvirinae* [1]. It is the infectious agent that causes Aujeszky’s disease (AD), also named pseudorabies, first described in 1902 by Aladár Aujeszky [2]. AD presents respiratory problems, neurological disorders, and abortions mainly in swine, leading to significant economic losses in the affected farms [3]. In swine, PRV can establish a lifelong latent infection in the nervous system [4,5], an ability that makes eradication of the virus very difficult [6]. In addition to swine, a large variety of mammalian species can serve as terminal hosts of PRV, including various ruminants, carnivores, and rodents [7,8,9,10,11]. Moreover, although humans are not natural hosts for PRV, 13 sporadic cases of encephalitis and endophthalmitis linked to this virus in humans have been reported since 1950 [7,12,13,14,15]. Whether PRV is able to infect humans is still controversial. Recently, a PRV strain was firstly isolated from a patient suffering from acute human encephalitis [16]. This strain revealed close phylogenetic and etiological characteristics similar to variant strains of Chinese PRV, suggesting the existing risk of PRV transmission from pigs to humans.

AD is considered to be eradicated from several European countries, the USA, Canada, Mexico, and New Zealand, where the attenuated pseudorabies virus vaccine strain Bartha-K61 was preventively applied in massive control campaign. This vaccine, which lacks the genes that encode the glycoproteins gE and gI and harbors other independent mitigating defects was extensively used, but alternative eradication programs used either naturally attenuated or recombinant gG or gC-deleted vaccines [4,17,18]. Vaccination was prohibited in countries that managed to eradicate the disease [17,19]. However, China was not able to fully eradicate the disease, and numerous outbreaks of AD have been detected in Chinese swine farms [6,18,20,21]. Due to the emergence of new PRV variants for which the Bartha-K61 vaccine does not protect [6,22], new vaccines and treatments are now needed for the control of AD.

Currently, there are no antiviral drugs to treat PRV infections; thus, new control measures are urgently needed. One of the most effective treatment options against viruses from the *Alphaherpesvirinae* subfamily (which includes herpes simplex virus type 1 [HSV-1]) is the use of nucleoside analogues like acyclovir (ACV), a guanosine analogue that interferes with viral DNA replication [23,24]. Nonetheless, ACV shows several drawbacks, such a short half-life and carcinogenic and embryotoxic effects, and the appearance of PRV strains with mutations in the gene encoding viral thymidine kinase (where ACV acts) could confer these strains resistance to this compound [23,25]. Furthermore, the use of this antiviral does not protect against the reactivation of the virus in the event that it has established latency [24] and the possibility of co-infection in pigs with PRV and a great variety of both bacterial and viral pathogens makes treatment difficult [6].

An alternative that remains under study is valproic acid (VPA) [26], a branched, short-chain fatty acid used in the treatment of seizure disorders [27]. Numerous studies have proven that VPA interferes with the infectious cycle of several enveloped viruses, including herpesviruses, flaviviruses, arenaviruses, poxviruses, picornaviruses, rhabdoviruses, and coronaviruses, among others [28,29,30,31,32,33]. However, VPA shows hepatotoxic and teratogenic activity [34], and thus to reduce these deleterious effects, derivatives of VPA have been tested. For example, valpromide (VPD) and valnoctamide (VCD) are amide derivatives of VPA that differ slightly in structure (Figure 1) and present less toxic effects compared to VPA [26,35,36,37].

VPD is a prodrug used in the treatment of neurological disorders such as convulsive, non-convulsive, and myoclonic epilepsies, and it has a shorter half-life and fewer adverse effects than VPA [38,39,40]. After oral or intravenous administration in humans, VPD is rapidly biotransformed through metabolic hydrolysis into its corresponding acid (90% of the total VPD dose) [41]. Furthermore, pharmacological trials showed a potency of the clinical effect of VPD three times greater than that of VPA [42]. Previous studies have shown the ability of VPD to prevent the replication of HSV-1 in glial cells [38], and its antiviral effects were observed at clinically permitted concentrations, thus it may be a good substitute for nucleoside analogues in the treatment against other alphaherpesviruses [38].

In the present study, we tested the potential antiviral activity of VPD against PRV infection in the neuroblastoma Neuro-2a cell line and PK15 swine cell line. VPD was shown to be non-cytotoxic at a specific range of concentrations, and its antiviral activity against PRV in vitro suggests that VPD should be considered to be an antiherpetic drug.

## 2. Materials and Methods

### 2.1. Cell Cultures

The PK15 swine cell line was derived from kidney epithelial cells of an adult pig [43], and was generously provided by Dr. Yolanda Revilla (CBMSO, Madrid, Spain). Neuro-2a cells (N2a, CCL-131 American Type Culture Collection, Manassas, VA, USA) were originated from mouse neuroblasts with neuronal and amoeboid stem cell morphology isolated from brain tissue. The HOG cell line was established from a surgically removed human oligodendroglioma [44] and kindly provided by Dr. A. T. Campagnoni (University of California, Los Angeles, CA, USA). The Vero cell line was derived from the kidney of an adult African green monkey and was kindly provided by Dr. Enrique Tabarés (UAM, Madrid, Spain).

All cell lines were grown in culture medium (CM) containing low-glucose Dulbecco’s modified Eagle medium (DMEM) (Life Technologies, Paisley, UK) supplemented with 5% (for PK15, HOG and Vero) or 10% (for N2a) fetal bovine serum (FBS), and penicillin (50 U/mL) and streptomycin (50 μg/mL) at 37 °C in a humidified atmosphere of 5% CO_2_.

### 2.2. Viruses

Wild type PRV strain NIA-3 and recombinant strain PRV-XGF [45] were kindly provided by Dr. Enrique Tabarés (UAM, Madrid, Spain). PRV-XGF was obtained by replacing the gene encoding the PRV gG glycoprotein (glycoprotein that is not part of the virion as it is secreted into the medium by infected cells, being not essential for virulence) with the EGFP gene.

NIA-3 and PRV-XGF were propagated and titrated on N2a and PK15 cells respectively by endpoint dilution assay.

### 2.3. Antibodies and Reagents

ACV (PHR1254), VPD (V3640), low-glucose DMEM, Phalloidin TRIT-C and FBS were purchased from Sigma Chemical Co. (Merck Chemicals, Darmstadt, Germany). Mowiol and DAPI (268298) were from Calbiochem (Merck Chemicals, Darmstadt, Germany). For Western blots, polyclonal rabbit anti-GFP from ChromoTek, (Planegg-Martinsried, Germany) and monoclonal mouse anti-β-actin-peroxidase (A7854) from Sigma Aldrich (Merck Chemicals, Darmstadt, Germany) were used as primary antibodies, with horseradish peroxidase-conjugated secondary anti-rabbit IgG antibody from Thermofisher (Waltham, MA, USA) used as secondary antibody.

### 2.4. Viral Infections

For viral infections, cells were plated in culture dishes and infected or mock-infected with NIA-3 or PRV-XGF in CM supplemented with 2% FBS or serum-free medium, respectively at an m.o.i. of 0.5. After 1 h of viral adsorption, cells were washed with PBS and left in CM supplemented with 10% FBS for 24 h at 37 °C in a humidified atmosphere of 5% CO_2_.

### 2.5. Endpoint Dilution Assay

Sub-confluent monolayers of N2a or PK15 cells were plated in 96-well tissue culture dishes and cultured in CM supplemented with 10% or 5% FBS, respectively. Serial dilutions (10^−1^ to 10^−9^) of NIA-3 or PRV-XGF were prepared and inoculated onto replicate cell cultures. Cells were then incubated at 37 °C in a humidified atmosphere containing 5% CO_2_ for 48 h. Finally, the 50% tissue culture infectious dose (TCID_50_) per ml was determined, considering the final dilution that showed cytopathic effect and calculated using the Reed and Muench method [46].

### 2.6. Immunofluorescence Microscopy

PK15 cells grown on glass coverslips were fixed in 4% paraformaldehyde for 20 min and rinsed with PBS. Cells were then permeabilized with 0.2% Triton X-100, rinsed, and incubated for 30 min at room temperature with 3% bovine serum albumin in PBS (blocking buffer). Coverslips were stained with phalloidin TRIT-C for 20 min in the dark and then washed 5 times with PBS, and nuclei were stained with DAPI for 10 min. After thorough washing, coverslips were mounted in Mowiol. Images were obtained using an LSM 710 Inverted Confocal Microscope (Zeiss, Vienna, Austria), and processing of confocal images was performed using the Fiji-ImageJ software (version Image J 1.53c).

### 2.7. Cell Viability Assay

The cytotoxic effects of ACV and VPD in HOG, N2a, PK15, and Vero cell lines were analyzed by the MTT method (Promega, Cell Titer 96^®^ Non-Radioactive Cell Proliferation Assay). Non-confluent monolayers of cells plated in 96-well tissue culture dishes and cultured in CM were incubated for 24 h with different concentrations of ACV (between 0 and 100 μM) and VPD (between 0 and 4 mM). Four replicates were carried out for each concentration. Then, cells were incubated with a final concentration of 0.5 mg/mL of MTT in a humidified atmosphere for 4 h, at which point formazan crystals were solubilized in 10% SDS in 0.01 M HCl. The resulting coloured solution was quantified using the scanning multiwell spectrophotometer iMark^TM^ Microplate Reader (BioRad, Hercules, CA, USA), measuring the absorbance of formazan at 595 nm. The readouts obtained from MTT assay were further normalized to the value of untreated cells where the viability value was set to 100%.

### 2.8. Virucidal Effect of VPD against PRV

The ability of VPD to directly interfere with PRV infection of PK15 cells was evaluated. PRV-XGF was mixed with different concentrations of VPD (0, 15, 20 or 40 mM) and incubated for 1 h at 37 °C in a humidified 5% CO_2_ incubator. Subsequently, each mixture was used to inoculate confluent monolayers of PK15 cells in 24-well plates, resulting in a dilution of VPD to a final concentration of 0, 1.5, 2.5, or 4 mM, and a viral m.o.i. of 0.5. Three replicates were made for each concentration. After 1 h of adsorption, cells were washed twice with PBS and incubated for 24 h in CM supplemented with 10% FBS.

Two additional controls were added to this assay. In control P2, PK15 cells were infected with PRV-XGF at an m.o.i. of 0.5 in a standard way, such that the virus could enter the cells normally and infect them. After 1 h of adsorption, cells were washed with PBS and VPD at a concentration of 1.5 mM was added for 24 h. In control P3, PK15 cells were infected with PRV-XGF at an m.o.i. of 0.5 in the presence of 1.5 mM of VPD during the adsorption of the virus, with no pre-incubation of the virus and drug. After 1 h of adsorption, cells were washed and cultured in CM supplemented with 10% FBS for 24 h. Results were analyzed by Western blot, and viral titer (TCID_50_/mL) was quantified by titration endpoint dilution assay in PK15 cells, as described previously.

### 2.9. Antiviral Effect of ACV and VPD in PK15 Cells Infected with PRV-XGF

Confluent monolayers of N2a or PK15 cells cultured in CM were plated in 24-well tissue culture dishes and pre-treated or mock-treated for 30 min with ACV (10 μM), VPD (0.5, 1 or 1.5 mM) or a mix of both drugs (ACV 10 μM + VPD 1.5 mM). Six biological replicates were made for each concentration. N2a or PK15 cells mock-treated or treated with either drug were infected with NIA-3 (N2a and PK15) or PRV-XGF (PK15) at an m.o.i. of 0.5, and after 1 h of viral adsorption, cells were washed with PBS and maintained as a post-treatment in CM supplemented with 10% FBS with either ACV (10 μM), VPD (0.5, 1 or 1.5 mM) or the mix (ACV 10 μM + VPD 1.5 mM) for 24 h at 37 °C in a humidified 5% CO_2_ incubator. The effect of VPD on viral infection was evaluated either by Western blot or quantification of viral production (TCID_50_/mL) by endpoint dilution assay in N2a and PK15 cells as described previously.

### 2.10. Immunoblot

Cell samples from virucidal and antiviral assays were lysed and analyzed using a Bradford assay to equalize the protein load. Then, cells were subjected at 24 h post-infection (hpi) to SDS-PAGE in 10% acrylamide gels under non-reducing conditions and transferred to Merck Millipore Immobilon-P membranes (Merck, Darmstadt, Germany). After blocking with 5% non-fat dry milk and 0.05% Tween 20 in PBS, blots were incubated overnight at 4 °C with the appropriate primary antibody. After several washes with 0.05% Tween 20 in PBS, blots were incubated for 1 h with secondary antibody coupled to horseradish peroxidase, washed extensively, and developed using an enhanced chemiluminescence Western blotting kit, the ECL^TM^ Western Blotting Detection Reagents, (GE Healthcare, Chicago, IL, USA).

### 2.11. Statistics

Results are expressed as the mean ± standard deviation. Data were subjected to Mann–Whitney U-tests (using Prism software v8.0.1, GraphPad software, Inc., San Diego, CA, USA) to determine significant differences between groups, and *p* < 0.01 was considered statistically significant.

## 3. Results

### 3.1. VPD and ACV Are Non-Toxic in Neither N2a nor PK15 Cells at Clinically Relevant Concentrations

To study the toxicity of VPD and ACV in both N2a and PK15 cell line, cells were cultured for 24 h in culture medium (CM) in the presence of different concentrations of each compound. This assay was also performed in Vero and HOG cells as controls. Cells treated with 1.5 mM VPD or 10 μM ACV reached approximately 80% viability. These were considered to be the highest non-toxic concentrations of the compounds, which were subsequently used in the rest of this study (Figure 2). Both drugs administered at the same time (VPD 1.5 mM + ACV 10 μM) for 24 h maintained viability above 80% in N2a and PK15 cell lines (results not shown).

### 3.2. Infection of PK15 Cells with PRV

To confirm whether the PK15 cell line was susceptible to recombinant PRV-XGF infection, cells were plated in 24-well tissue culture dishes on round coverslips and infected or mock-infected with PRV-XGF at an m.o.i. of 0.5. After 1 h of viral adsorption, cells were washed and cultured in CM for 24 h at 37 °C in a humidified atmosphere of 5% CO_2_. Bright-field microscopy images (Figure 3A) showed remarkable cytopathic effect (CPE) in vivo in PRV-infected PK15 cells compared to the mock-infected controls. Fluorescence microscopy also demonstrated the efficient infectivity of PRV-XGF in fixed PK15 cells (Figure 3B).

### 3.3. VPD Shows No Virucidal Effect against PRV

To evaluate whether VPD had a direct virucidal effect against PRV, PRV-XGF was mixed with VPD at different concentrations and the mix was incubated for 1 h. Then, PK15 cells were infected with this mixture at an m.o.i. of 0.5 for 1 h, and after double washing with PBS, cells were incubated for 24 h in CM supplemented with 10% FBS. Immunoblot analysis revealed a single unique band corresponding to viral GFP (Figure 4A), and the presence of this band at equal intensity regardless of VPD concentration indicates that the drug did not directly block or inactivate the virus. These results were further supported by the infectious titer data: there was no significant decrease in viral titer in any of the samples treated with different concentrations of VPD, or controls P2 and P3, in comparison to untreated but infected PK15 cells (Figure 4B).

### 3.4. Antiviral Effect of VPD and ACV on PRV Infection of PK15 Cells

To evaluate whether VPD had an antiviral effect against PRV, we incubated the cell lines with VPD at different concentrations for 30 min, prior to infection. Then, we infected PK15 cells with PRV-XGF and NIA-3 and N2a cells with NIA-3 at an m.o.i. of 0.5 for 24 h. Viral production was determined by endpoint dilution assay, showing a drastic decrease in viral production in VPD-treated cells compared to mock treatment. The number of infectious viral particles in VPD-treated cells at all tested concentrations decreased significantly (*p* < 0.01 at 24 hpi) compared to the non-treated cells (Figure 5A). 1.5 mM VPD was the concentration that elicited a greater decrease in the infection. The decrease of PRV-XGF infection against PK15 cells was also demonstrated by immunoblot analysis, where the 25 kDa band corresponding to viral GFP showed less intensity when VPD concentration was increased (Figure 5B). Relative to 10 µM ACV, VPD at 1.5 mM showed a similar level of antiviral activity. Finally, when cells were treated at the same time with ACV and VPD, no significant decrease of the infection was observed.

## 4. Discussion

The current lack of a specific treatment for AD and the emergence of new strains of PRV that are able to overcome protection from existing vaccines make it essential to find effective antivirals against PRV [6]. The nucleoside analogues that have been traditionally used as anti-herpetic drugs, including ACV or ganciclovir, exhibit carcinogenic, embryotoxic, and teratogenic activities [23,25], and an increasing number of immunocompromised patients are resistant to ACV [47,48]. Various studies have confirmed the in vitro antiviral effects of VPD against some herpesviruses such as HSV-1 [29,39], Epstein-Barr virus (EBV) [49,50], and cytomegalovirus (CMV) [51]. PRV shares many characteristics with HSV-1 [52], as they both belong to the subfamily *Alphaherpesvirinae*; thus, based on this, we hypothesized that VPD would also have an antiviral effect against PRV [26].

VPD has been shown to block EBV reactivation in two relevant Burkitt lymphoma cell lines [49], thus it was decisive to test whether VPD could similarly inhibit PRV reactivation in target neuronal cells. Therefore, this study was carried out not only on the PK15 swine cell line, but also on Neuro-2a mouse neuroblastoma cells (PRV is able to infect rodents).

On the other hand, it was crucial to test the efficacy of VPD against PRV infection with a wild-type viral strain such as NIA-3. As this strain is not labeled with any reporter protein, once the antiviral activity of VPD was demonstrated in such strain, the following experiments were performed with recombinant PRV-XGF (developed by replacing the gene encoding the PRV gG glycoprotein with the EGFP gene). PRV gG has chemokine-binding properties and plays a crucial role in immune evasion [45], but it has been shown that this glycoprotein gG does not interfere with the efficacy of VPD.

The virucidal assay demonstrated that VPD at the tested concentrations does not block or inactivate PRV-XGF directly, suggesting that it may act as a prodrug, needing to be processed by cellular machinery to reach its active form. Thus, pre-treatment of cells with VPD was necessary [39]; this requirement for VPD metabolization has been empirically reported for infection of HOG cells with HSV-1 [29,38]. Either VPD or ACV were added to the cells at the same time as the virus (Figure 4, control P3) and no reduction in the infection was detected. Immediately after adsorption of the virus, post-treatment is required to avoid a loss of effect. The virucidal effect of ACV was not assayed because it is a prodrug that must first be transformed by cellular kinases [24,53].

Regarding antiviral activity, VPD managed to reduce the infection of PRV by 90% at the concentration of 1.5 mM in all cell lines tested. Besides, a dose-dependent antiviral effect of VPD was confirmed in this study. The combination of acyclovir with other antiviral drugs has shown synergistic or additive effects against HSV, VZV, and CMV [54]. Nevertheless, combination of VPD and ACV did not show significant differences between the drugs tested separately.

The antiviral mechanism and kinetics of VPD are not yet fully understood. Previous studies with HSV-1 indicated that VPD and its precursor VPA affect the initial steps of virus entry [29,38]. VPA acts at several different molecular levels: it potentiates the inhibitory activity of the neurotransmitter gamma amino butyrate [55], inhibits NMDA receptor-mediated excitatory transmission, Na^+^ channels [56], T-Type Ca^2+^ and K^+^ channels, and acts as a histone deacetylase inhibitor (HDAC_i_) [27]. VPA and VPD also alter lipid metabolism and formation of cell membranes; therefore, it is possible that these drugs affect the process of acquisition of lipid envelope by enveloped viruses, leading to mature viral particles with lower stability [57].

In terms of teratogenicity, VPD does not show any such effect in pregnant rats, mice, swine, or dogs [37,58]. However, since VPD is biotransformed into teratogenic VPA in humans, this drug is still contraindicated in women of childbearing age or pregnant women, since it may exhibit the same teratogenic effects [26]. Nonetheless, VPD could still be useful for treatment in animal species where VPD is only partially metabolized, such as in swine, dogs, or mice [37,59,60].

In conclusion, VPD was proven to have an inhibitory effect on PRV infection comparable to ACV in swine PK15 and neuroblastoma N2a cells, thus it would be interesting to further investigate VPD as a suitable alternative to nucleoside analogues as an antiherpetic drug against AD. VPD managed to reduce the infection by one order of magnitude. This remarkable reduction compensates for the need to administer the drug as a pretreatment (prior to cellular infection). Considering that VPD is already licensed for clinical use by the EMA (European Medicines Agency), the time required to approve its new use for the treatment of alphaherpesvirus infections should be shorter than for all-new drugs. Further research is needed to unravel the detailed mechanism responsible for the antiherpetic activity of VPD.

## Figures and Tables

**Figure 1 viruses-13-02522-f001:**
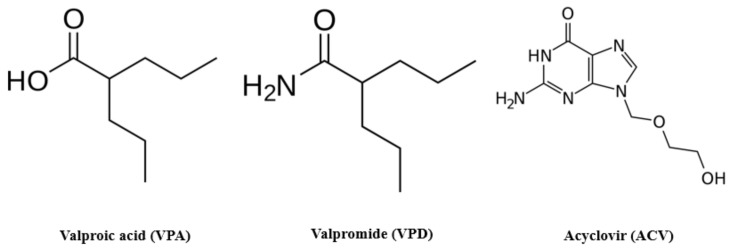
**Molecular structure of valproic acid (VPA), valpromide (VPD), and acyclovir (ACV).** VPA differs from its amidic derivative VPD in the acidic core. ACV is one of the most popular drugs chosen for anti-herpetic treatments.

**Figure 2 viruses-13-02522-f002:**
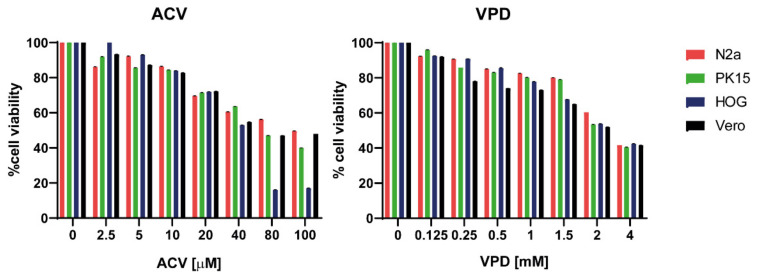
**Viability of N2a, PK15, HOG, and Vero cells exposed to acyclovir (ACV) and valpromide (VPD).** Cells were cultured in culture medium (CM) and treated or mock-treated for 24 h with different concentrations of VPD (between 0.125 mM and 4 mM) or ACV (between 2.5 μM and 100 μM). Cell viability was measured by MTT tetrazolium salt assay, and calculated as the percentage of viability compared to untreated cells; columns represent the mean viability ± SD (*n* = 6) after exposure to the drugs.

**Figure 3 viruses-13-02522-f003:**
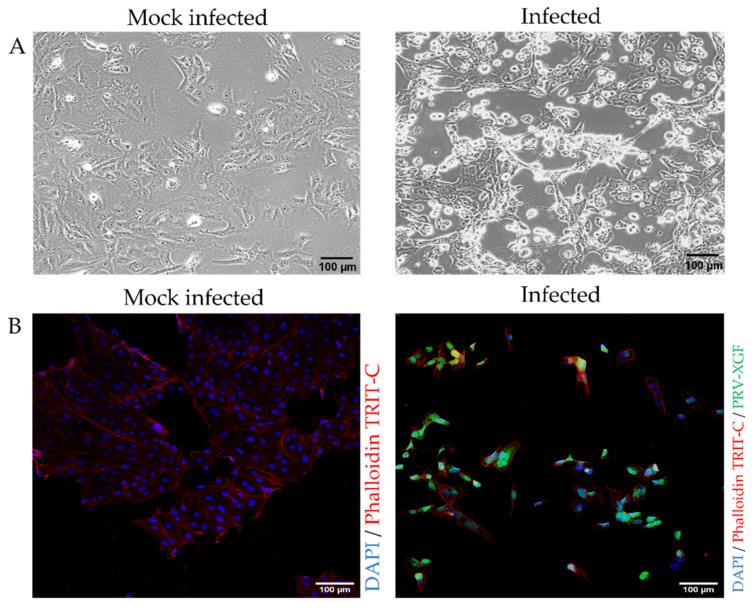
**Susceptibility of PK15 cells to recombinant PRV-XGF infection.** PK15 cells cultured in CM were mock-infected or infected with PRV-XGF at an m.o.i. of 0.5 for 24 h. (**A**) Bright field images show CPE in vivo of PK15 cells infected with PRV-XGF at 24 hpi compared to mock-infected cells. (**B**) Fluorescence microscopy images of PK15 cells infected with PRV-XGF show GFP signal corresponding to viral infection. Mock-infected cells are also shown. For better visualization, all cells were stained with DAPI and phalloidin TRIT-C.

**Figure 4 viruses-13-02522-f004:**
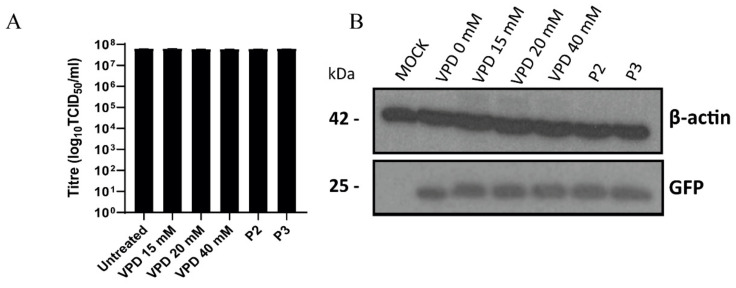
**Valpromide (VPD) shows no virucidal effect against PRV.** PRV-XGF was mixed with VPD at different concentrations (0, 15, 20, or 40 mM) for 1 h and then each mix at a final viral m.o.i of 0.5 was added to PK15 cells for 1 h of adsorption. For control P2, infected cells at an m.o.i of 0.5 after 1 h of adsorption were incubated for 24 h in the presence of 1.5 mM VPD. For control P3, virus and 1.5 mM VPD were simultaneously added to cells for 1 h of adsorption. Infected cells were washed with PBS and then cultured in medium supplemented with 10% FBS for 24 h at 37 °C in a humidified 5% CO_2_ incubator. (**A**) Progeny virus was titrated in PK15 cells at 24 h post-infection to determine the 50% tissue culture infective dose (TCID_50_)/mL. The graph shows the mean ± S.D. (*n* = 3) viral production in PK15 cells. (**B**) Viral GFP protein was analyzed by immunoblot in total cell lysates. Equal numbers of cells were lysed, processed using a Bradford assay to equalize the protein load, subjected to SDS-PAGE, and analyzed using immunoblotting with anti-GFP antibody. β-actin was used as protein loading control.

**Figure 5 viruses-13-02522-f005:**
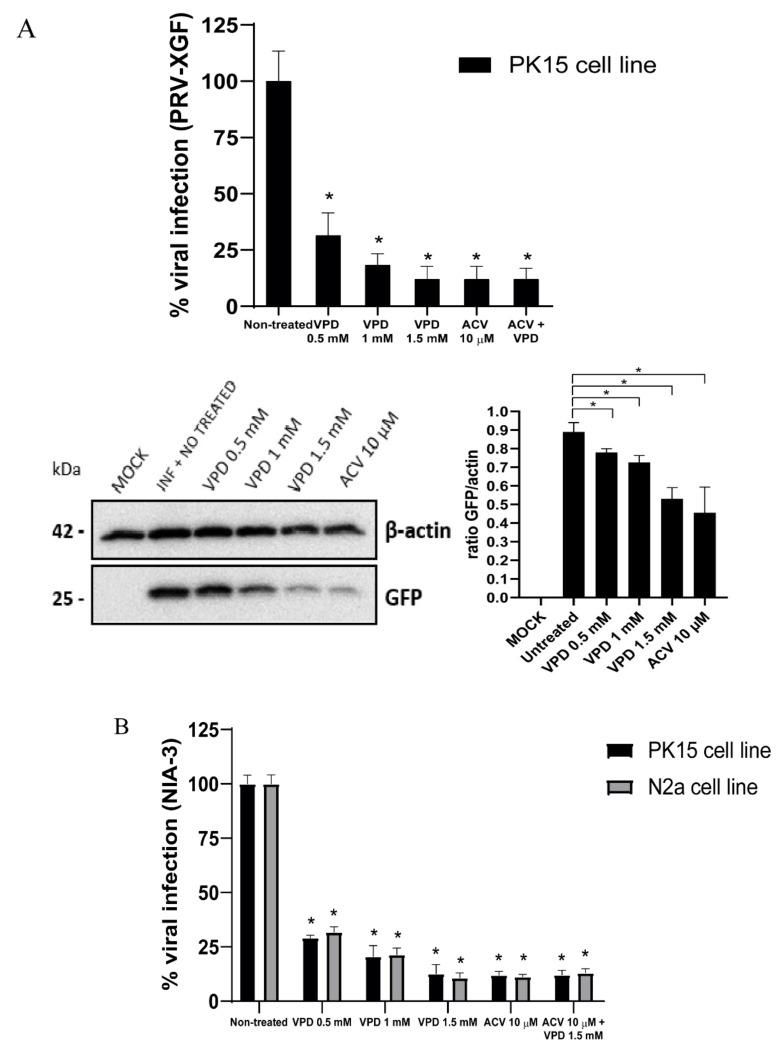
**Valpromide (VPD) and acyclovir (ACV) significantly decrease PRV infection in N2a and PK15 cells.** N2a and PK15 cells cultured in 24-well dishes were pre-treated for 30 min with VPD (at concentrations 0, 0.5, 1, or 1.5 mM), ACV (10 µM), or a mix of both drugs (ACV 10 µM + VPD 1.5 mM) prior to infection with NIA-3 or PRV-XGF, at an m.o.i. of 0.5 for 1 h. After that, cells were washed with PBS and cultured in medium supplemented with 10% FBS and either VPD (at the concentrations 0, 0.5, 1 or 1.5 mM), ACV (10 µM) or the mix (ACV 10 µM + VPD 1.5 mM) for 24 h at 37 °C in a humidified 5% CO_2_ incubator. (**A**) PRV-XGF progeny virus was titrated in PK15 cells at 24 h post-infection to determine the 50% tissue culture infective dose (TCID_50_)/mL. The graph shows the mean percentage of viral infection ± S.D. (*n* = 6); statistical comparison is done between non treated cells and the rest of the conditions*: *p* < 0.01. Viral GFP protein of PRV-XGF was analyzed by immunoblot in total PK15 cell lysates. Equal numbers of cells were lysed, processed using a Bradford assay to equalize the protein load, subjected to SDS-PAGE, and analyzed using immunoblotting with anti-GFP antibody. β-actin was used as protein loading control. Values of immunoblot quantification are reported as the mean ± S.D. (*n* = 2), *: *p* < 0.05. (**B**) NIA-3 progeny virus was titrated in PK15 or N2a cells at 24 h post-infection to determine the 50% tissue culture infective dose (TCID_50_)/mL. The graph shows the mean percentage of viral infection ± S.D. (*n* = 6); a statistical comparison was done between non treated cells and the rest of the conditions*: *p* < 0.01.

## Data Availability

Not applicable.

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
