# Peer review of "The Valproic Acid Derivative Valpromide Inhibits Pseudorabies Virus Infection in Swine Epithelial and Mouse Neuroblastoma Cell Lines"

_viruses, 2021, doi:10.3390/v13122522_

Round 1
Reviewer 1 Report
This manuscript reported the screening of two candidates inhibiting PRV, which have some clinical significance. Overall the study is well-designed and the English writting is also OK. However, it should be better if an in vivo tests can be included. The effectiveness of the candiates should be tested in proper animal models such as mouse models. As far as I know, many candidates show good effects on inhibiting a bacterial or a viral pathogen in vitro, but their effect is not as good as expect in vivo models. Therefore, I think in vivo tests are really important.
In addition, the quality of several figures, e.g., Figure 4A, is poor, and should be improved.
Author Response
Dear reviewer,
Thank you very much for your comments. We strongly believe that they have been very helpful in improving the quality of the manuscript.
The study we have conducted is a preliminary assay in which we indeed tested the in vitro activity of valpromide in various cell lines, and against two types of PRV strains.
Of course, our purpose is in the near future to continue this study with suitable animal models, such as rodents and, if possible, pigs. We consider the study to be a preliminary study prior to a future approach to the assay in models.
We have in mind to perform an in vivo study with the collaboration of Cinta Prieto from the Universidad Complutense de Madrid, researcher of the Department of Animal Health.
With respect to figure 4A, we have improved blot image quality.
Thank you.
Reviewer 2 Report
Dear editor, Dear authors,
The manuscript has greatly improved. The authors took into account all my comments and suggestions from the first round of revision. I am very satisfied with the final result. Congratulations to the authors!!!
Author Response
Dear reviewer,
Thank you very much for your comments. We really believe that they have helped us to improve the paper and increase its quality.
Round 2
Reviewer 1 Report
I do not have any further comments
This manuscript is a resubmission of an earlier submission. The following is a list of the peer review reports and author responses from that submission.
Round 1
Reviewer 1 Report
The study from Andreu and colleagues investigates the antiviral effect of valpromide against pseudorabies virus (PRV) infection in swine cells. The manuscript is well-written and structured. However, the manuscript presents several major issues that are addressed below:
Major comments:
1° Lines 40-43: I would still develop the part on PRV infection in humans. Please mention exactly how many human cases have been reported so far (See Laval & Enquist, 2020). Most cases were reported in China. The high incidence of suspected human PRV infections identified might to be related to the high prevalence of PRV in swine in China and repeated exposure to infected animal tissues. Please also indicate that PRV DNA was mostly detected in human tissue samples and the infectivity of the virus was never determined by virus titration. There is no clear evidence that the virus is infectious in humans. This information is very important for the readers and for the rationale of this study.
2° Lines 55-56: What is the added value to use valpromide if ACV is already the most effective antiviral drug against herpesvirus infection? Is ACV as effective for both PRV and HSV1 infections? Please comment on the rationale of your study here. Lines 268-279 should be included in the introduction. Is VPD for use in pigs or humans, or both? Is VPD cheaper than ACV? Some information to these questions are already provided in the manuscript but should be clearly stated in a paragraph in the introduction.
3° Lines 50-52: You talked about the emergence of PRV new variants and the fact that the Bartha vaccine fails to protect against them. Can antivirals protect against variants as well? Please be more specific here.
4° VPD has been shown to block EBV reactivation in two relevant Burkitt lymphoma cell lines (Gorres et al., 2016). It is important to test whether VPD can similarly inhibit PRV reactivation in target neuronal cells. This study needs to include a neuronal cell type. As PRV infects rodents, it might be more convenient to use rodent primary neuronal cell cultures (e.g. superior cervical ganglion) or neuroblastoma cells (e.g., neuro2A) than swine neuronal cells.
5° The choice of the PRV strain used is not clear. It is crucial to test the efficacy of VPD against PRV infection with a wild-type viral strain (e.g., PRV-Becker). Why did the authors choose to first use a PRV mutant (gG deleted)? PRV gG has chemokine-binding properties and plays a crucial role in immune evasion. The gG might potentially interfere with the efficacy of VPD in vivo. The in vitro experiments need to be repeated with a wild-type PRV strain.
6° Lines 240- 251: Please include representative pictures showing a decreased PRV antigen staining in VPD-treated cells compared to untreated cells.
7° Lines 295-297: VPA is an HDAC inhibitor shown to activate herpesvirus transcription in target cells. Does VPD work as an HDAC inhibitor as well? If so, it might be a problem and therefore the efficacy of VPD as an antiviral to PRV infection should be tested in target cells.
8° Lines 299-304: Do you mean here that VPD is not an alternative to ACV treatment in humans? This is in contradiction with lines 63-68. Would it be a cost-effective treatment in pigs?
Minor comments:
9° Lines 31 and 54: Please be consistent with the writing (italic or not).
10° Lines 129-136: Please mention staining against PRV glycoproteins here.
11° Figure 1: Change Y axis with ticks every 25 % and up to 100% and not more.
12° Line 248: Extra space in the sentence.
13° Line 265. Correct “4. Discussion”.
Reviewer 2 Report
Overall, there is very little novelty in this publication. It has previously been described that VPA has anti-herpesvirus activity. The performed experiments are simple and don’t add additional information nor comparative information. Similarly, there are discrepancies between experiments for figure 4 and 5, also in comparison to previously published work that need addressed. More details or additional questions worth asking see below. However, recommended further work:
So lots more to be done on this piece of research:
- Proper dilution curves, calculation of IC50, IC90
- Comparing effects in different cell lines (also dependent on aim of the study)
- Combination effect of ACV and VPD combination treatment
- Possibly comparison to other antivirals - ribavirin e.g. in combination treatments
- Time of addition experiments
- Tying up results and checking stats
--
In the introduction, the authors switch between human impact and porcine disease of PRV. What is the aim of this study to use VPA for? Treatment in humans or treatments in animals? This is unclear throughout. If human use is considered, surely VPA efficacy and proper IC50, IC90 inhibitory curves and calculations should be performed on all susceptible cell lines representing model infection (Vero), human infection (HOG), and porcine infection (PK-15) cells.
ACV is relatively expensive and therefore has been primarily used in companion animals. Cost factors of treatment versus vaccines should be discussed. On the other hand VPA treatment is very dangerous during treatment in pregnancy, this again should be highlighted as a potential negative in the use of VPA (or VPD) in animal husbandry.
One of the reports cited highlights two cases of PRV infections in humans where VPA treatment was part of the regime to treat seizures yet the treatment did not indicate any impact on the virus (Fan et al. J Neurovirol. 2020;26:556–564. doi: 10.1007/s13365-020-00855-y). The authors should do further literature research into this area as VPA is used frequently for treatment of human PRV infection symptoms and indication of its antiviral effect may be observed.
The experiments performed here are very narrow and many obvious experiments towards PRV treatment have not been done. Why did the authors not perform any combination treatments since the traditional treatment of ACV is likely to occur in combination with VPAs. Combination matrices and calculation of additive versus synergistic effects should be performed. Other drugs known to show synergistic effects should be included, such as ribavirin, known to potentiate the effect of ACV for PRV so the question here is whether this cheap drug also potentiates VPA treatment (in direct comparison to ribavirin/ACV).
And lastly, I have got a treatment difference question on Figures 4 and 5. It does not make sense to me at present how control P2, with an incubation at 15mM VPD for 24h and during inoculation would not have an impact contrast to a 30min pre-treatment. Whilst I’d expect some impact of a pre-treatment, if the VPD acts on entry, surely there should still be a decrease in virus replication upon 10x VPD (contrast to the 1.5mM in figure 5). If this is truly so black and white then a time of addition experiment must be performed and a known virucidal added as a control for figure 4. Comparing this to previous work this also does not make sense since addition even at 2hpi still showed complete inhibition, 4hpi 50% inhibition, and 6hpi 25% inhibition in HSV infection.
Methods:
Statistics:
Mann-Whitney test does not seem appropriate to analyse this data. There’s no ranking involved but rather a control versus different treatments. The inappropriateness of this should be visible to the authors as well in that for example in figure 5A, results are clearly highly significant with a 3x and above decrease in released virus. However, all results “only” show p<0.05, unless the authors did not bother to display higher significance appropriately.
Western Blot quantification:
First, the methods of Western blotting are vague – several washes is not specific, nor is washing extensively. However, most importantly, the method of quantification must be described in more details. Was chemiluminescence measured by film or using a chemiluminescene reader, if so, which one?
The reduction of GFP in figure 5B appears almost too little compared to virus reduction in 5 A, however, figure 5A is difficult to read due to the lack of a log scaling. I am therefore tempted to assume that these blots were scanned and quantified using a count of dark or light pixels and the usual problem of overexposure of the high intensity bands in the MOCK TREATED, infected (hopefully this is not an untreated infection control!) leads to a decrease in intensity. But all assumptions since the methods are not described.
Results / Discussion
Figure 2:
Why were HOG and Vero cells chosen as controls for cell viability? HOG cells appear particularly sensitive to VPD treatment whereas PK15 cells appear comparatively robust. It is clear that Vero cells are highly susceptible to PRV infection, why were they not used in parallel to test VPD treatment? There appears to be an immediate drop in cell viability even at the lowest concentration, which is then level for quite some concentrations. Do the “0” concentration cells here represent untreated or mock treated cells? Also, treatment in PK15 cells appears concentration dependent but not in HOG or Vero cells, which is a bit odd.
Figure 3:
Figure 3 is a nice visualisation but the claims are to show that PK15 cells are susceptible to PRV infection. This has already been demonstrated in the past in several publications. The authors claim in B that they show in these images that PRV-XGF GFP signal corresponds to infection. Whilst I believe the authors that this is correct, surely to prove this a counterstaining with an anti-PRV protein antibody would be the way to prove this. Also to see whether some viruses loose GFP over passaging.
Figure 4:
Given that the VPD was not removed from the virus stocks prior to inoculation of the cells this graph does not only represent virucidal activity but also effects on entry. The authors should highlight this. Also, the concentrations used have previously been described as highly cytotoxic. I am actually surprised that there was not a simple effect of cell viability on virus replication after a 1h “shock” at this concentration.
Figure 5:
Could the authors display the Figure A with a log scale and reassess the statistics, as discussed in the methods section. A split axis should be considered if the authors want to highlight the difference of the 0.5mM VPD. Currently I cannot quite tell whether VPD and ACV have similar or differing activity due to the scale and the lack of information on statistical significance and the way this is displayed.